# Effects of the COVID-19 Pandemic on Physical Function of Community-Dwelling People with Disabilities in Japan

**DOI:** 10.3390/ijerph191912599

**Published:** 2022-10-02

**Authors:** Takayuki Kamimoto, Michiyuki Kawakami, Towa Morita, Yuta Miyazaki, Nanako Hijikata, Tomonori Akimoto, Masahiro Tsujikawa, Kaoru Honaga, Kanjiro Suzuki, Kunitsugu Kondo, Tetsuya Tsuji

**Affiliations:** 1Department of Rehabilitation Medicine, Keio University School of Medicine, Tokyo 160-8582, Japan; 2Department of Rehabilitation Medicine, Tokyo Bay Rehabilitation Hospital, Chiba 275-0026, Japan; 3Department of Physical Rehabilitation, National Center of Neurology and Psychiatry, Tokyo 187-8551, Japan; 4Department of Rehabilitation Medicine, National Cancer Center Hospital East, Chiba 277-8577, Japan; 5Department of Rehabilitation Medicine, Juntendo University Graduate School of Medicine, Tokyo 113-8421, Japan; 6Department of Rehabilitation Medicine, Waseda Clinic, Miyazaki 880-0933, Japan

**Keywords:** rehabilitation, COVID-19, cerebrovascular disease, musculoskeletal disease

## Abstract

In 2020, COVID-19 spread throughout the world, and international measures such as travel bans, quarantines, and increased social distancing were implemented. In Japan, the number of infected people increased, and a state of emergency was declared from 16 April to 25 May 2020. Such a change in physical activity could lead to a decline in physical function in people with disabilities. A retrospective study was conducted to determine the impact of the pandemic on the physical function of disabled persons living in the community. Data were collected at four points in time: two points before the declaration of the state of emergency was issued and two points after the declaration period had ended. Time series data of physical function at four points in time were compared for 241 people with disabilities. The mean age was 72.39 years; 157 had stroke, 59 musculoskeletal disease, and 26 other diseases. Overall, there was a long-term decrease in walking speed (*p* < 0.001) and a worsening of the Timed Up-and-Go (TUG) score (*p* < 0.001) after the period of the state of emergency. The TUG score worsened only in the group with a walking speed of 1.0 m/s or less before the state of emergency (*p* = 0.064), suggesting that this group was more susceptible.

## 1. Introduction

On 31 December 2019, China reported to the World Health Organization an outbreak of a novel coronavirus causing pneumonia in adults in Wuhan, Hubei Province. The spread of the coronavirus has forced people to change their lifestyles, requiring them to take measures against infectious diseases, such as social distancing. Internationally, measures such as travel bans, isolation, and increased social distance were recommended and implemented [1,2]. As the number of infected people gradually increased in Japan, the Japanese government issued an emergency declaration on 7 April 2020. Although it was not legally binding, it had a certain effect and reduced the number of infections; it was lifted in all prefectures on 25 May 2020 [3]. It was recommended to avoid the 3 Cs, places where COVID-19 spreads more easily: 1. Crowded places, 2. Close-contact settings, and 3. Confined and enclosed spaces. In turn, people’s living arrangements were gradually reduced; it is known that, during the current pandemic, time spent performing moderate to vigorous physical activity decreased by 60% among young adults, and time spent sedentary increased by 42% [4].

There have been reports of the effects of this lifestyle on older adults living in the community. Lockdowns were accompanied by decreased physical activity, disordered eating, stress, and altered sleep patterns, and the risk of sarcopenia increased, especially in older adults, which may lead to the progression of multiple lifestyle-related diseases, along with impacting quality of life and exercise capacity [5]. In previous studies, weight gain, decreased grip strength, and decreased walking speed were observed in older adults after lockdowns compared with before [6,7].

Frailty is increasing in all countries and is a major cause of functional decline and early mortality in older people [8]. People with frailty may experience a decline in physical function, especially as their range of activity decreases. Japanese older adults reported a decrease in physical activity in April 2020, and subjective lower limb muscle weakness was more common in the group with frailty [9].

People with disabilities have been differentially affected by COVID-19 because of three factors: the increased risk of poor outcomes from the disease; reduced access to routine health care and rehabilitation; and the adverse social impacts of efforts to reduce disability [10]. Frailty occurs more frequently in people with disabilities such as stroke, Parkinson’s disease, post hip fracture, and heart failure [11,12,13,14]. Therefore, it has been reported that lockdowns and decreased outings limit the amount of activity in people with disabilities, affecting them in ways such as worsening balance ability and increasing the risk of falls [15,16,17]. However, these reports were limited to a limited number of diseases, such as Parkinson’s disease and multiple sclerosis, they were short-term evaluations, with only a 2-month pre- and post-lockdown evaluation, and the number of participants was limited to a dozen or so people.

In addition, there have been several reports of physical function assessed over time during hospitalization and after discharge in COVID-19 patients [18,19]. On the other hand, there are a limited number of papers examining the impact of changes in social activity due to COVID-19 on physical functioning in a large number of community-dwelling people with disabilities. In this study, the actual impact of the coronavirus pandemic on the physical function of persons with disabilities living in a community in Japan was investigated. This study shows that lifestyle-altering events such as COVID-19 are a risk for motor function loss in people with disabilities. This will clarify the need for and appropriate timing of efforts to prevent motor function loss in people with disabilities during the next lifestyle-altering event (including unknown infectious diseases) that may come along again.

## 2. Materials and Methods

### 2.1. Inclusion and Exclusion Criteria

The inclusion criteria were: (1) participants must have started or continued to use daycare from 1 September to 31 December 2019; and (2) participants must have continued to use daycare through 31 August 2020. Since daycare is a service available through Long-Term Care Insurance (LTCI), which is described below (Section 2.2), all participants were at least 40 years old, and there were no restrictions with regard to their diseases.

To assess the impact of the spread of the coronavirus over time, the following four time periods were evaluated: T1, from 1 September 2019 to 31 December 2019; T2, from 1 January 2020 to 6 April 2020, (a state of emergency was declared in Japan from 7 April 2020 to 25 May 2020); T3, from 26 May 2020 to 31 August 2020; and T4, from 1 September 2020 to 31 December 2020. (Figure 1) All periods lasted 4 months. During the state of emergency, the Japanese government recommended avoiding the 3 Cs, refraining from going out, and working from home.

The exclusion criterion was two or more missing data points in the four T1–T4 periods.

An opt-out consent process was used because of the retrospective nature of the study. The study was conducted according to the guidelines of the Declaration of Helsinki and approved by the Ethics Committee of Tokyo Bay Rehabilitation Hospital (No. 284, approved 12 January 2022).

### 2.2. Daycare and Long-Term Care Insurance (LTCI)

The study was conducted at a single facility providing rehabilitation to maintain and improve the function of community dwelling people with disabilities. LTCI is a public system that began in 2000, and those over 40 years of age are obligated to pay premiums; half of the premiums are paid by the public, and the rest by the national government, prefectures, and municipalities. LTCI services are provided to insured persons who are certified for support or care requirements according to their care needs and certification assessment. The insurance benefits include in-home services and services at facilities, including long-term care welfare facilities, long-term care health facilities, and long-term care medical facilities. Dependent older adults can select and use provided facility, in-home, or community-based services according to their care needs, and care managers are actively involved in care plans and service arrangements [20]. One of the LTCI services is daycare, where people with disabilities and older adults in Japan receive regular rehabilitation with the goal of recovering and maintaining physical and mental functions and independence in daily life. At this daycare facility, once or twice a week, physical therapists, occupational therapists, and speech therapists formulate exercise programs suited to each client and conduct individual rehabilitation in addition to circuit training-style exercise.

### 2.3. Physical Data

Periodic assessments of physical functions were conducted about every three to six months.

The 10 m walk test is a method of assessing walking ability that has been used to evaluate a variety of conditions, including stroke, hip fracture, and spinal cord injury [21,22,23]. It is also a reliable method for assessing walking ability in elderly persons [24,25]. It measures speed during a 10 m walk, and currently, there are no clear rules regarding acceleration and deceleration intervals before and after the 10 m segment. In the present study, a 3 m acceleration interval and a 3 m deceleration interval before and after, respectively, were set, and the 10 m walking speed during both intervals was measured with a digital stopwatch [26]. The time taken to walk a 10 m segment at a comfortable speed was measured twice, and the faster value was used to determine walking speed. During the test, participants were permitted to use walking devices and/or an ankle-foot orthosis. The criterion for walking speed was determined to be 1.0 m/s (meters per second). The most well-known and widely accepted standard for measuring physical frailty is the “Cardiovascular Health Study criteria” [27]. In Japan, there is a revised Japanese version of the Cardiovascular Health Study criteria (revised J-CHS criteria) [28]. In the revised J-CHS criteria, one of the criteria for frailty is walking speed under 1.0 m/s. In addition, walking speed slowness (<1.0 m/s) is useful for screening for frailty in cancer patients over 65 years of age [29].

The TUG, a modification of the Get-Up-and-Go test, was developed primarily to assess basic motor skills in frail older adults, and it is reliable and valid [30,31]. The TUG consisted of a series of movements: getting up from a 40-cm-high armchair, walking 3 m, turning around, and sitting down again. The task was performed and measured twice, and the faster value was used for the TUG score.

The modified Rankin Scale (mRS) is a simple scale for assessing physical function and ADL and has been shown to be reliable and valid in patients with stroke [32]. It has six levels, with scores from 0 to 5; those with scores of 3 or higher are those requiring assistance with some ADL.

### 2.4. Statistical Analyses

Statistical analyses were performed using IBM-SPSS Statistics (version 25, 2017, IBM Corporation, Armonk, NY, USA). Parametric data are expressed as means ± standard deviation (SD) or 95% confidence interval. Baseline differences in the mean walking speed of T1–T2 (T1–T2 mean walking speed) by sex, disease, and mRS scores were examined using the *t*-test and one-factor analysis of variance (ANOVA). To compare and analyze the effects of the time period, sex, and the T1–T2 mean walking speed (Fast group (≥1.0 m/s) and Slow group (<1.0 m/s)), ANOVA with the mixed-effects model for repeated measures (MMRM) was performed, with the Bonferroni test as the post hoc test. In all cases, *p* values less than 0.05 were considered significant.

## 3. Results

A total of 329 people used daycare during the time period T1 and continued to use it through 31 August 2020. After applying exclusion criteria, 241 daycare users were eligible for this study.

Table 1 shows the demographic data: age 72.39 ± 10.21 years; male 130 (53.9%), female 111 (46.1%); 157 (65.1%) had stroke; 58 (24.1%) had a musculoskeletal disease; and 26 (10.8%) had other diseases (Parkinson’s disease: 10, cancer: 4, spinal cord injury: 3, heart failure: 3, chronic kidney disease: 2, allergic granulomatous angiitis: 1, encephalopathy: 1, arteriosclerosis obliterans: 1, and interstitial pneumonia: 1). The number with an mRS score of 2 or lower was 94, and the number with an mRS score of 3 or greater was 147. The mean mRS score was 2.45 ± 0.93. The mean walking speed at T1–T2 was not significantly different by sex, disease, or mRS score. The number in the Fast group was 94, and the number in the Slow group was 147.

First, the differences in the mean values of walking speed and TUG at different time periods were examined. The mean walking speeds were: T1 0.87 m/s; T2: 0.86 m/s; T3 0.78 m/s; and T4 0.79 m/s. The results of MMRM showed a significant difference by the time period (F (3, 633) = 46.57 (*p* < 0.001)). Bonferroni’s post hoc test showed significant differences in T1–T3 (*p* < 0.001), T1–T4 (*p* < 0.001), T2–T3 (*p* < 0.001), and T2–T4 (*p* < 0.001). (Figure 2a) The mean TUG scores at each time period were: T1 15.52 sec; T2 15.54 sec; T3: 16.66 sec; and T4 17.02 sec. The results of MMRM showed a significant difference by time period (F (3, 628) = 8.26 (*p* < 0.001)). Bonferroni’s post hoc test showed significant differences in T1–T3 (*p* = 0.016), T1–T4 (*p* < 0.001), T2–T3 (*p* = 0.020), and T2–T4 (*p* < 0.001) (Figure 2b).

Second, to examine the effect of sex, ANOVA with MMRM was performed. Significant differences were obtained by time period (*p* < 0.001). However, there was no significant difference by sex (*p* = 0.091), with no interaction effect between sex and the time period (*p* = 0.646).

Third, to examine the effect of walking speed, ANOVA with MMRM was performed. The result of MMRM showed significant differences by time period, the walking speed group (Fast group and Slow group), the interaction effect between the time period and the walking speed group (*p* < 0.001), and the interaction effect between the time period and the TUG score (*p* < 0.001). The mean walking speeds were significantly different for T1–T3 (*p* < 0.001), T1–T4 (*p* < 0.001), T2–T3 (*p* < 0.001), and T2–T4 (*p* < 0.001) in the Fast group, and for T1–T3 (*p* = 0.006), T1–T4 (*p* < 0.001), T2–T3 (*p* = 0.044), and T2–T4 (*p* = 0.009) in the Slow group. (Figure 3a) The mean TUG scores were significantly different for T1–T3 (*p* = 0.031), T1–T4 (*p* < 0.001), and T2–T4 (*p* = 0.007) in the Slow group, whereas there was no significant difference by time period in the Fast group (*p* = 0.064) (Figure 3b).

## 4. Discussion

This study involved Japanese daycare users with disabilities whose mRS scores were between 1 and 4, and it evaluated the changes in gait disorder, one of the “disabilities”, over time before and after COVID-19 pandemic. The strengths and appealing points of this study are that the data were obtained through periodic evaluations of persons with disabilities living in the community, which is valuable, because previous reports were limited. The present study suggested two conclusions: first, the ability to walk (walking speed and TUG) deteriorated in homebound people with disabilities following the COVID-19 epidemic; and second, the effects of the coronavirus persisted even after the national government’s request to refrain from leaving home was lifted.

People with disabilities had reduced physical function after the period of restriction. This is considered appropriate, considering that previous studies have shown that activity restrictions due to the pandemic reduced physical functions such as walking speed in elderly persons [7] and balance ability in Parkinson’s disease patients [15], which seems to be appropriate given these results. Though there are generally multiple factors that may be responsible for the decrease in walking speed, in the present study, one of the major causes may have been the decreased activity level. The declaration of a state of emergency and the adoption of new lifestyles, such as avoiding the aforementioned three Cs, have certainly reduced the amount of activity of the population. In fact, it has been reported that the amount of physical activity of older adults in Japan decreased in April compared to January 2020 [9]. It is possible that the unintended decrease in activity due to self-restraint, i.e., reduced walking, causes muscle weakness in the extremities [33]. As a result, physical function, which was maintained or improved between T1 and T2, may have been exacerbated by the self-restraint. Although it is difficult to identify a single cause, because the TUG contains a complex set of abilities, such as walking speed, muscle strength, and static/dynamic balance capacity, the TUG has been shown to be useful in predicting fall risk in older adults [34], and these results suggest an increased risk of falls.

In the present study, physical function continued to decline even after the self-restraint was lifted. This is likely due to the fact that physical fitness is difficult to regain once it has declined in people with disabilities. It is also possible that people’s lifestyles have changed, and they are not as active as they once were, even after the period of self-restraint ended. Globally, people in many countries experienced a large decrease in steps in March 2020 after the declaration of the coronavirus pandemic, but steps have gradually recovered since then, whereas in Japan, the decrease in steps has persisted [35]. Even though activity levels declined among older adults in Japan during the lifting of the emergency declaration [20], after the lifting of the emergency declaration, the overall activity level recovered, but activity levels in the socially inactive group who lived alone did not recover. It is highly possible that the socially inactive group consisting of people with disability was more likely to be socially inactive, and it was difficult for them to recover their activity level. It has also been noted that there are regional differences [36], and since the present study was conducted in the Tokyo metropolitan area of Japan, the baseline activity level was originally higher than in regional cities, making the area more susceptible to a decrease in activity level due to the declaration of a state of emergency.

In the sub-analysis, the mean walking speed before the declaration of emergency (T1–T2) was divided into two groups using 1.0 m/s as the cut-off. The results showed a clear difference in TUG scores between the Fast group and the Slow group. In terms of walking speed, the two groups showed a similar deterioration in walking speed regardless of the baseline walking speed. On the other hand, TUG scores did not change before and after the emergency declaration in the group that originally had a walking speed faster than 1.0 m/s. However, a decrease in TUG scores was observed in the slow group, which had a walking speed slower than 1.0 m/s. The Fast group, on the other hand, showed a simple decrease in comfortable walking speed, but not to the point of worsening balance ability or fall risk, since a TUG score of 13.5 or higher is considered a predictor of falls in community-dwelling older adults [37]. The Slow group had a mean TUG score of 13.5 or higher at T1; therefore, the slow group had a very high risk for falls. The Slow group is a high-risk group for the development of frailty [38], and the presence of frailty may be associated with worsening balance and lower limb muscle strength. The presence of frailty is correlated with worsening balance and lower limb muscle weakness [39], and the TUG, which includes multiple factors such as muscle strength and balance capacity, may have been particularly susceptible to the effects of frailty.

There are some potential limitations of the present study. First, the study was conducted at only one institution, and thus the results must be generalized with caution. It is also unclear how the pandemic influenced the lifestyle and activity of community-dwelling people with disability because there are no data involving assessment of the trends in physical activity.

## 5. Conclusions

In this study, the time series data analysis suggests that activity limitations associated with the COVID-19 pandemic affected the physical function and gait ability of people with disabilities living in the community, indicating that the effects of the pandemic are long-lasting and protracted. The TUG score worsened only in the Slow group before the state of emergency, suggesting that this group was more susceptible. Lifestyle-altering events such as the COVID-19 pandemic have been shown to be a risk factor for physical decline in people with disabilities, and the negative effects on physical function are prolonged. Therefore, early and sustained preventive interventions will be necessary for future events of any kind. The creation of a system that can provide safe and effective training using new methods, such as remote rehabilitation, will be essential for the future of society. In particular, it is important to create an environment in which continuous rehabilitation and maintenance of activity can be ensured even in emergency situations, especially for those with disabilities who are vulnerable.

## Figures and Tables

**Figure 1 ijerph-19-12599-f001:**
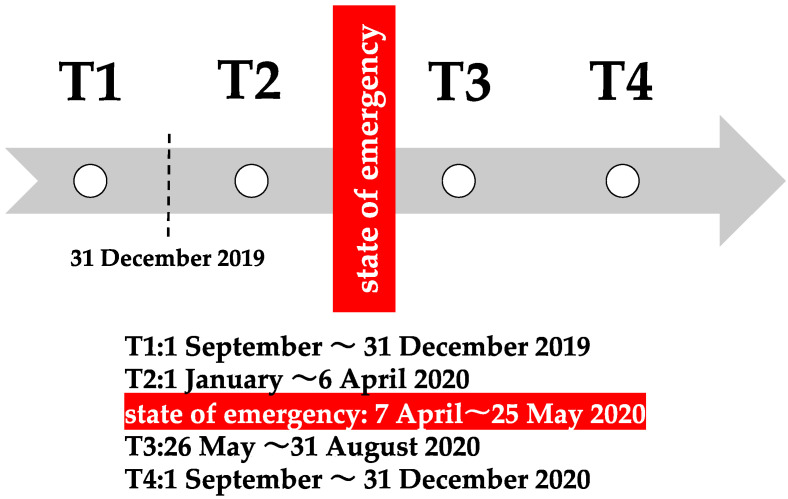
Timeline for the periods of analysis. On 31 December 2019, an outbreak of a novel coronavirus was reported. To evaluate the effect of the pandemic, the following four-time points were evaluated: T1, T2, T3, and T4.

**Figure 2 ijerph-19-12599-f002:**
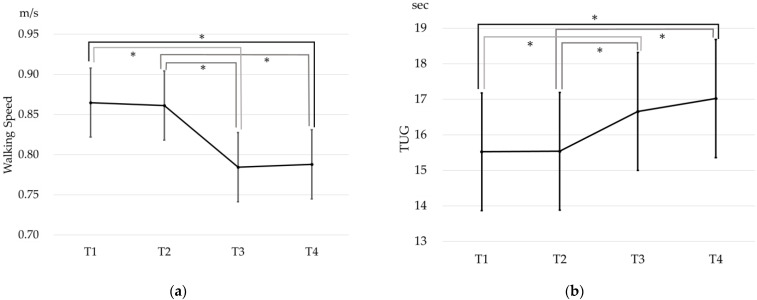
The physical data at each time period. (**a**) The mean walking speed at each time period (95% confidence interval); (**b**) the mean TUG score at each time period (95% confidence interval). * Bonferroni’s post hoc test: Significant difference (*p* < 0.05) TUG: Timed-Up-and-Go test.

**Figure 3 ijerph-19-12599-f003:**
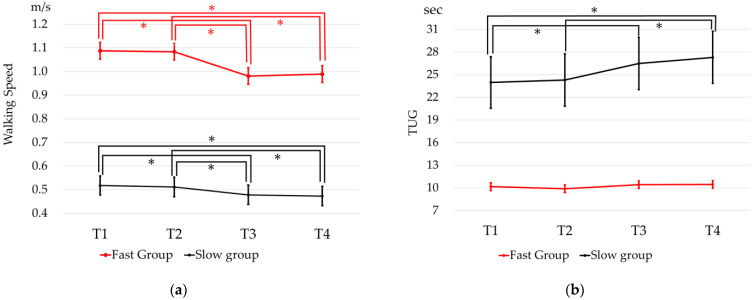
The physical data of each group (Fast group and Slow group) at each time period. Slow group, T1–T2 mean walking speed < 1.0 m/s; Fast group, T1–T2 mean walking speed ≥ 1.0 m/s. (**a**) The mean walking speed of each group (Fast group and Slow group) at each time period (95% confidence interval); (**b**) the mean Timed-Up-and-Go test (TUG) score of each group (Fast group and Slow group) at each time period (95% confidence interval). * Bonferroni’s post hoc test: significant difference (*p* < 0.05).

**Table 1 ijerph-19-12599-t001:** Demographic data of the 241 daycare users. mRS: modified Rankin Scale, SD: standard deviation, a: *t*-test, b: One-factor ANOVA.

		Mean	SD	n		%	T1–T2 Mean Walking Speed (m/s)		*p*-Value
							Mean	SD	
Age (years)		72.39	10.21				0.89	0.34	
Sex	male			130		53.9%	0.89	0.34	0.191 ^a^
	female			111		46.1%	0.83	0.35	
Disease	stroke			157		65.1%	0.85	0.37	0.303 ^b^
	musculoskeletal disease			58		24.1%	0.92	0.26	
	others			26		10.8%	0.82	0.30	
mRS	1			58	94	39.0%	0.85	0.35	0.739 ^a^
	2			36					
	3			130	147	61.0%	0.87	0.34	
	4			17					

## Data Availability

Not applicable.

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
