# Peer review of "Effects of the COVID-19 Pandemic on Physical Function of Community-Dwelling People with Disabilities in Japan"

_ijerph, 2022, doi:10.3390/ijerph191912599_

Round 1

Reviewer 1 Report

This is an interesting study which aims to investigate the impact of the COVID-19 pandemic on physical function of persons with disabilities living in a community in Japan. The study appears methodologically well conducted and the manuscript accurately written.

Comments

1) In my opinion the title should also include “in Japan”.  Since restrictions, lockdown and state of emergency have been different in countries around the world, it would be better to specify even in the title that the study is limited to a specific country, namely Japan.

2) Please, can the authors clarify or better describe the types of disabilities/impairments (i.e. physical or mental functioning) that the population under study was affected by?

3) In the statistical analysis section it is reported that data are expressed as means ± standard deviation but in the figures, they appear as mean and 95% confidence interval. This should be added for completeness.

4) As a last comment, figures should show also T4 point. 

Author Response

Comments

1) In my opinion the title should also include “in Japan”.  Since restrictions, lockdown and state of emergency have been different in countries around the world, it would be better to specify even in the title that the study is limited to a specific country, namely Japan.

Thank you for your comments. The wording of the title has been corrected as follows:

-“Effects of the COVID-19 pandemic on physical function of community-dwelling people with disabilities in Japan”

2) Please, can the authors clarify or better describe the types of disabilities/impairments (i.e. physical or mental functioning) that the population under study was affected by?

To describe impairments/disabilities that affected the persons with disabilities, we have added the following in the “Discussion”:

-“This study was conducted on Japanese daycare users with disabilities whose mRS were between 1 and 4, and it evaluated the changes in gait disorder, one of the “disability”, over time before and after COVID-19 pandemic.

3) In the statistical analysis section it is reported that data are expressed as means ± standard deviation but in the figures, they appear as mean and 95% confidence interval. This should be added for completeness.

We have added the following in “Statistical Analyses”:

-“mean±SD or 95 percent confidence interval.”

4) As a last comment, figures should show also T4 point. 

-It seems that when we pasted the figures, they became misaligned, and T4 disappeared; this has been corrected.

Reviewer 2 Report

I find it an interesting study. I would like them to add more arguments to support the research problem (they say "Few studies of the impact of the COVID-19 pandemic on physical functions, such as the ability to walk, have been conducted in people with disabilities", but what is the problem of this?)

Make the objective more clear. And that this be answered in the conclusions

Add in the discussion the strengths of your study, what is new and the practical usefulness of what you found.

Author Response

I find it an interesting study. I would like them to add more arguments to support the research problem (they say "Few studies of the impact of the COVID-19 pandemic on physical functions, such as the ability to walk, have been conducted in people with disabilities", but what is the problem of this?)

Make the objective more clear. And that this be answered in the conclusions

Add in the discussion the strengths of your study, what is new and the practical usefulness of what you found.

Thank you for your comments. We have revised and expanded the “Introduction” and “Conclusion” to clarify the purpose and social significance of our study, as follows:

Introduction:

- “This study reveals that lifestyle-altering events such as COVID-19 are a risk for motor function loss in people with disabilities. This will clarify the need for and appropriate timing of efforts to prevent motor function loss in people with disabilities during the next lifestyle-altering event (including unknown infectious diseases) that may come along again.”

Conclusion:

-“Lifestyle-altering events such as COVID-19 have been shown to be a risk for physical decline in people with disabilities, and the negative effects on physical function are prolonged. Therefore, early and sustained preventive intervention will be necessary for the next event of any kind.”

Reviewer 3 Report

The paper submitted for review deals with the effect on the physical function of community-dwelling persons with disabilities during the COVID-19 pandemic, which is meaningful for revising formal public health policies.

However, I have some doubts or suggestions, which, I hope, will improve the quality of your article.

Line 42-62: Since the paper focused on the person with disabilities, the literature review on the physical function of the elderly is too long. I suggest summarizing the two sections (Line 42-50 and Line 51-61) into one, add a section to introduce reference about the correlation between frailty and disabilities.

Line 79-80: To our knowledge, there are already some papers dealing with a similar problem that focused on the impact of the COVID-19 pandemic on physical functions (1,2). Suggestions to include in this manuscript and to consider in terms of knowledge are already known. The contention of the authors of this manuscript that the topic was not previously discussed seems too exaggerated. If the author aimed to express that there were few studies focused on the effect on the person with disabilities, this sentence needs to be refined to avoid misunderstandings.

Line 79-82: This section lacked the significance and hypothesis of this research. Such as: what could this research do for public health, society, et al.

Line 88-91: “Each period included 4 to 5 months” : Were the different lengths of periods confounding variables in this research? I suggest adding this to the limitation.

Line 164-165: Since the subjects were all over 60 years of age, which means that they were all older adults. Should the title of the research be modified? e.g. Effects of the COVID-19 pandemic on physical function of the community-dwelling older adults with disabilities.

Figure 2, Figure 3: The research included 4 time points, while the figure only presented 3-time points. It is necessary to modify the four figures.

Line 271-279: The conclusion is general and does not highlight the results of the study. The effect on the group that originally had a walking speed slower than 1.0m/s was more serious than the group faster than 1.0m/s. This means that the pandemic was more likely to damage the health of the person with poor physical function. This was the highlight of the study in my opinion.

[1] Bertacchini L, Paneroni M, Comini L, Scalvini S, Vitacca M. Recovering of oxygenation, physical function and disability in patients with Covid-19. Monaldi Arch Chest Dis. 2021;91(4):10.4081/monaldi.2021.1817. Published 2021 Apr 9. doi:10.4081/monaldi.2021.1817

[2] Vieira AGDS, Pinto ACPN, Garcia BMSP, Eid RAC, Mól CG, Nawa RK. Telerehabilitation improves physical function and reduces dyspnoea in people with COVID-19 and post-COVID-19 conditions: a systematic review. J Physiother. 2022;68(2):90-98. doi:10.1016/j.jphys.2022.03.011

Author Response

The paper submitted for review deals with the effect on the physical function of community-dwelling persons with disabilities during the COVID-19 pandemic, which is meaningful for revising formal public health policies.

However, I have some doubts or suggestions, which, I hope, will improve the quality of your article.

Line 42-62: Since the paper focused on the person with disabilities, the literature review on the physical function of the elderly is too long. I suggest summarizing the two sections (Line 42-50 and Line 51-61) into one, add a section to introduce reference about the correlation between frailty and disabilities.

Thank you for your comments. We have summarized the two sections, as suggested, and we have added text focusing on the correlation between frailty and disability in the “Introduction”, as follows:

- Lockdown was accompanied by decreased physical activity, disordered eating, stress, and altered sleep patterns, and the risk of sarcopenia increased, especially in older people, which may lead to the progression of multiple lifestyle-related diseases, along with the impact on quality of life and exercise capacity. [5] In Brazilian older women, weight gain and decreased grip strength compared to pre-lockdown levels were observed, suggesting sarcopenia. [6] In Italian older adults, there was a decrease in walking speed and grip strength after lockdown. [7] Restriction of mobility has a significant impact on the physical, psychological, and social functioning of elderly persons. [8]

Japan is one of the few super-aged societies in the world. In fact, as of 2019, 28.4% of the Japanese population was over 65 years old, and by 2040-2050, the rate is expected to reach 40%. [9] Frailty is increasing in all countries and is a major cause of functional decline and early mortality in elderly persons. [10] A meta-analysis in Japan reported that 53.6% of older adults aged 65 years or older in the community were frail or pre-frail. [11] Such individuals may experience a decline in physical function, especially as their range of activity decreases. Japanese older adults reported a decrease in physical activity in April compared to January 2020, and a survey of older adults after the corona outbreak showed that subjective lower limb muscle weakness was more common in the frail group. [12] It has been reported that subjective lower limb muscle weakness was more significant in the group with frailty in a survey of older adults after a coronal event. [13]

People with disabilities have been differentially affected by COVID-19 because of three factors: the increased risk of poor outcomes from the disease; reduced access to routine health care and rehabilitation; and the adverse social impacts of efforts to reduce disability. [14]…

-“Lockdown was accompanied by decreased physical activity, disordered eating, stress, and altered sleep patterns, and the risk of sarcopenia increased, especially in older poeple. In previous studies, a weight gain, decreased a grip strength, and decreased a walking speed were observed in older adults compared with those before lockdown. [6][7]

           Frailty is increasing in all countries and is a major cause of functional decline and early mortality in older people. [8] People with frailty may experience a decline in physical function, especially as their range of activity decreases. Japanese older adults reported a decrease in physical activity in April 2020, and subjective lower limb muscle weakness was more common in the frailty group. [9]

People with disabilities have been differentially affected by COVID-19 because of three factors: the increased risk of poor outcomes from the disease; reduced access to routine health care and rehabilitation; and the adverse social impacts of efforts to reduce disability. [10] “Frailty occurs more frequently in people with disabilities such as stroke, Parkinson’s disease, post hip fracture, and heart failure. [11][12][13][14] Therefore, it has been reported that lockdowns and outings limit the amount of activity in people with disabilities, affecting them in ways such as worsening balance ability and increasing the risk of falls. [15][16][17] However, these reports were limited to a limited number of diseases, such as Parkinson's disease and multiple sclerosis, and were short-term evaluations, with only a 2-month pre- and post-lockdown evaluation, and the number of participants was limited to a dozen or so people.”

Line 79-80: To our knowledge, there are already some papers dealing with a similar problem that focused on the impact of the COVID-19 pandemic on physical functions (1,2). Suggestions to include in this manuscript and to consider in terms of knowledge are already known. The contention of the authors of this manuscript that the topic was not previously discussed seems too exaggerated. If the author aimed to express that there were few studies focused on the effect on the person with disabilities, this sentence needs to be refined to avoid misunderstandings.

[1] Bertacchini L, Paneroni M, Comini L, Scalvini S, Vitacca M. Recovering of oxygenation, physical function and disability in patients with Covid-19. Monaldi Arch Chest Dis. 2021;91(4):10.4081/monaldi.2021.1817. Published 2021 Apr 9. doi:10.4081/monaldi.2021.1817

[2] Vieira AGDS, Pinto ACPN, Garcia BMSP, Eid RAC, Mól CG, Nawa RK. Telerehabilitation improves physical function and reduces dyspnoea in people with COVID-19 and post-COVID-19 conditions: a systematic review. J Physiother. 2022;68(2):90-98. doi:10.1016/j.jphys.2022.03.011

We have cited the literature on rehabilitation after recovery from COVID-19 infection, as suggested, and though there is some literature on COVID-19 rehabilitation, we believe that the literature on long-term, sequential evaluation of changes in physical function associated with the COVID-19 epidemic in individuals who were disabled before the epidemic of COVID-19 infection is valuable, though limited. We believe that ours is a valuable study because there is limited literature that evaluates the changes in physical function over time in people who have been disabled since before the COVID-19 epidemic. We have revised the “Introduction”, as follows:

-“There have been several reports of assessing physical function over time during hospitalization and after discharge in COVID-19 patients. [18][19] On the other hand, there are a limited number of papers examining the impact of changes in social activity due to COVID-19 on physical functioning among community-dwelling people with disabilities”

Line 79-82: This section lacked the significance and hypothesis of this research. Such as: what could this research do for public health, society, et al.

We have revised and expanded the “Introduction” and “Conclusion” to clarify the purpose and social significance of our study, as follows:

Introduction:

-“This study reveals that lifestyle-altering events such as COVID-19 are a risk for motor function loss in people with disabilities. This will clarify the need for and appropriate timing of efforts to prevent motor function loss in people with disabilities during the next lifestyle-altering event (including unknown infectious diseases) that may come along again.”

Conclusion:

-“Lifestyle-altering events such as COVID-19 have been shown to be a risk for physical decline in people with disabilities, and the negative effects on physical function are prolonged. Therefore, early and sustained preventive intervention will be necessary for the next event of any kind.”

Line 88-91: “Each period included 4 to 5 months” : Were the different lengths of periods confounding variables in this research? I suggest adding this to the limitation.

We have corrected the inclusion criteria appropriately to have equal 4-month periods, and we have changed the sentences in the “Materials and Methods”, as follows:

The inclusion criteria were starting the use of daycare as of December 31, 2019, and then continuing its use through August 31, 2020.

- “The inclusion criteria were: 1) people must had started or continued to use daycare from September 1 to December 31, 2019; 2) people must continue to use the daycare through August 31, 2020.”

- With the appropriate changes in participation criteria, the overall number of data changed from 254 to 241. The analysis was performed again accordingly. The results themselves and the conclusions were not affected.

Line 164-165: Since the subjects were all over 60 years of age, which means that they were all older adults. Should the title of the research be modified? e.g.Effects of the COVID-19 pandemic on physical function of the community-dwelling older adults with disabilities.

Since this study included people under the age of 60 (minimum age of 48) years, we think it would be better to refer to them as “people” rather than as “older adults”.

Figure 2, Figure 3: The research included 4 time points, while the figure only presented 3-time points. It is necessary to modify the four figures.

It seems that when we pasted the figures, they were misaligned, and T4 disappeared; we have now corrected them.

Line 271-279: The conclusion is general and does not highlight the results of the study. The effect on the group that originally had a walking speed slower than 1.0m/s was more serious than the group faster than 1.0m/s. This means that the pandemic was more likely to damage the health of the person with poor physical function. This was the highlight of the study in my opinion.

We have added the following in the “Conclusion”, as suggested:

-“The TUG score worsened only in the group with the Slow group before the state of emergency, suggesting that this group was more susceptible.”

Reviewer 4 Report

Dear Authors,

As you have stated in your manuscript “The strengths and appealing points of this study are that the data are valuable because they were obtained through periodic evaluations of disabled persons living in the community.” I found your work remarkable in this respect. My suggestions as a peer are listed below.

The title is “Effects of the COVID-19 pandemic on physical function of community-dwelling persons with disabilities” but in your conclusion you have stated that “COVID epidemic affected the walking ability and the risk of falling of elderly persons living in the community” Line 272. Please correct accordingly.

In your abstract it would be better if you give some basic statistical information like p values,  mean age of the study group, their disabilities etc.

L 33. …”the Japanese government issued an emergency declaration on April 16, 2020”. But in L89 “a state of emergency was declared in Japan from April 7..” please correct.

Please give more details about the demographics of your study group, details of your inclusion criteria.. like an age range or any disability type.. etc. If all the subjects attending day care were older than 65 (or 40 ??) please state it.

L165 .. had cerebrovascular disease?? How many of them were stroke survivors? You have one pneumonia case in your study group, pneumonia is a form of acute respiratory infection that affects the lungs, since the time interval of your research is more than a year then it will not be feasible for you to use this diagnosis. This patient might have chronic obstructive lung disease or asthma. I suggest you to check from hospital records.

You have evaluated Modified Rankin Scale (L147-150) but you have not given any values about this neither in the results section, nor in discussion.

The 'timed up and go' test (TUG) is a simple, quick and widely used clinical performance-based measure of lower extremity function, mobility and fall risk. You have not evaluated balance directly, I think it would be better to use “fall risk” instead of balance.

 Best regards

Author Response

As you have stated in your manuscript “The strengths and appealing points of this study are that the data are valuable because they were obtained through periodic evaluations of disabled persons living in the community.” I found your work remarkable in this respect. My suggestions as a peer are listed below.

The title is “Effects of the COVID-19 pandemic on physical function of community-dwelling persons with disabilities” but in your conclusion you have stated that “COVID epidemic affected the walking ability and the risk of falling of elderly persons living in the community” Line 272. Please correct accordingly.

Thank you for your comments. We have changed the “Conclusion” as follows:

the COVID epidemic affected the walking ability and the risk of falling of elderly persons living in the community

-“COVID-19 epidemic affected the physical function of gait ability of people with disabilities living in the community”

In your abstract it would be better if you give some basic statistical information like p values, mean age of the study group, their disabilities etc.

We have added the statistical information to the “Abstract”, as follows:

-“The mean age was 72.39 years; stroke:157; musculoskeletal disease:58; and others:26. Overall, there was a long-term decrease in walking speed (p<0.001) and a worsening of the Timed Up-and-Go (TUG) score (p<0.001) after the period of the state of emergency. The TUG score worsened only in the group with a walking speed of 1.0 m/s or less before the state of emergency (p=0.064),...”

L 33. …”the Japanese government issued an emergency declaration on April 16, 2020”. But in L89 “a state of emergency was declared in Japan from April 7..” please correct.

We have changed the “Introduction”, as follows:

As the number of infected people gradually increased in Japan, the Japanese government issued an emergency declaration on April 14, 2020.”

-“As the number of infected people gradually increased in Japan, the Japanese government issued an emergency declaration on April 7, 2020.”

Please give more details about the demographics of your study group, details of your inclusion criteria.. like an age range or any disability type.. etc. If all the subjects attending day care were older than 65 (or 40 ??) please state it.

To clarify the demographics data of the study groups, we have added the following to sentences in the “Inclusion and Exclusion Criteria”:

-“Since day care is a service available through Long-Term Care Insurance (LTCI), which is described below (2.2), everyone is at least 40 years old, and no restrictions were placed on the inclusion of patient's disease.”

As the supplement data; “The mean age of 329 daycare users was 73.39 years old.”

L165 .. had cerebrovascular disease?? How many of them were stroke survivors? You have one pneumonia case in your study group, pneumonia is a form of acute respiratory infection that affects the lungs, since the time interval of your research is more than a year then it will not be feasible for you to use this diagnosis. This patient might have chronic obstructive lung disease or asthma. I suggest you to check from hospital records.

 We have revised the method of classification of diseases in the “Results” and in “Table 1”. We have changed the text as follows in the “Results”:

182 (71.7%) had cerebrovascular disease; 11 (4.3%) had a musculoskeletal disease; and 11 (4.3%) had other diseases (cancer: 4, heart failure: 3, chronic kidney disease: 2, arterio-sclerosis obliterans: 1, and pneumonia: 1)

-“…; 166 (65.4%) had stroke; 61 (24.0%) had a musculoskeletal disease; and 27 (10.6%) had other diseases (Parkinson’s disease:10, spinal cord injury:4, cancer: 4, heart failure: 3, chronic kidney disease: 2, allergic granulomatosis angiitis:1, encephalopathy:1, arteriosclerosis obliterans: 1, and interstitial pneumonia: 1), …”

You have evaluated Modified Rankin Scale (L147-150) but you have not given any values about this neither in the results section, nor in discussion.

The modified Rankin Scale (mRS) was listed for the purpose of making the characteristics of the present group easier to understand. This study involved Japanese daycare users with disabilities with an mRS score of 2.45 and did not include groups with scores of 0 or 5; thus, it did not include people who were completely bedridden or healthy people. The results are described in the “Results”, and the interpretation of the mRS data was added:

“Result”

-“The mean score of mRS was 2.45±0.93.”

“Discussion”

-” This study was conducted on Japanese daycare users with disabilities whose mRS were between 1 and 4,…”

The 'timed up and go' test (TUG) is a simple, quick and widely used clinical performance-based measure of lower extremity function, mobility and fall risk. You have not evaluated balance directly, I think it would be better to use “fall risk” instead of balance.

We have changed the text in the “Discussion” as follows:

This study suggested two conclusions: first, the ability to walk and muscle strength for balance deteriorated in homebound patients with disabilities following the corona-virus epidemic;

-”This study suggested two conclusions: first, the ability to walk (walking speed and TUG) deteriorated in homebound people with disabilities following the COVID-19 epidemic.”

Round 2

Reviewer 3 Report

All my concerns have been addressed.